# Proteomics, Phosphoproteomics and Mirna Analysis of Circulating Extracellular Vesicles through Automated and High-Throughput Isolation

**DOI:** 10.3390/cells11132070

**Published:** 2022-06-29

**Authors:** Hao Zhang, Yu-Han Cai, Yajie Ding, Guiyuan Zhang, Yufeng Liu, Jie Sun, Yuchen Yang, Zhen Zhan, Anton Iliuk, Zhongze Gu, Yanhong Gu, W. Andy Tao

**Affiliations:** 1State Key Laboratory of Bioelectronics, National Demonstration Center for Experimental Biomedical Engineering Education, Southeast University, Nanjing 210096, China; zhanghaohao@seu.edu.cn (H.Z.); yhcai@seu.edu.cn (Y.-H.C.); yajie_ding@seu.edu.cn (Y.D.); 230218260@seu.edu.cn (G.Z.); 230208230@seu.edu.cn (Y.L.); 230189808@seu.edu.cn (J.S.); 220181815@seu.edu.cn (Z.Z.); gu@seu.edu.cn (Z.G.); 2EVLiXiR Biotech, Nanjing 210032, China; 3Bell Mountain Molecular MedTech Institute, Nanjing 210032, China; 4Department of Oncology, The First Affiliated Hospital of Nanjing Medical University, Jiangsu Province Hospital, Nanjing 210029, China; yyc@njmu.edu.cn (Y.Y.); guyhphd@163.com (Y.G.); 5Tymora Analytical Operations, West Lafayette, IN 47906, USA; anton.iliuk@tymora-analytical.com; 6Department of Chemistry and Biochemistry, Purdue University, West Lafayette, IN 47907, USA

**Keywords:** extracellular vesicles, miRNA, phosphoproteomics, mass spectrometry, prostate cancer

## Abstract

Extracellular vesicles (EVs) play an important role in the diagnosis and treatment of diseases because of their rich molecular contents involved in intercellular communication, regulation, and other functions. With increasing efforts to move the field of EVs to clinical applications, the lack of a practical EV isolation method from circulating biofluids with high throughput and good reproducibility has become one of the biggest barriers. Here, we introduce a magnetic bead-based EV enrichment approach (EVrich) for automated and high-throughput processing of urine samples. Parallel enrichments can be performed in 96-well plates for downstream cargo analysis, including EV characterization, miRNA, proteomics, and phosphoproteomics analysis. We applied the instrument to a cohort of clinical urine samples to achieve reproducible identification of an average of 17,000 unique EV peptides and an average of 2800 EV proteins in each 1 mL urine sample. Quantitative phosphoproteomics revealed 186 unique phosphopeptides corresponding to 48 proteins that were significantly elevated in prostate cancer patients. Among them, multiple phosphoproteins were previously reported to associate with prostate cancer. Together, EVrich represents a universal, scalable, and simple platform for EV isolation, enabling downstream EV cargo analyses for a broad range of research and clinical applications.

## 1. Introduction

Extracellular vesicles (EVs) are secreted membranous nanoparticles containing functional molecules, such as nucleic acids, proteins, and metabolites, with diameters of 30 nm to 1000 nm. Most studies indicated that different cells could secrete EVs under normal or abnormal physiological conditions [1]. Packed with their active signaling molecules, EVs are utilized by cells as a significant system for cell-to-cell communication [2,3]. By transporting nucleic acids, proteins, lipids, amino acids, and metabolites, EVs play an important role in facilitating remote signal transmission in an organism, which is vital to the multi-organ collaboration and maintenance of normal physiological functions [4,5]. Many physiological processes have been discovered to be related to the EVs [6,7,8,9]. For example, Grizzle et al. found that tumor-derived vesicles could bear immunosuppressive molecules, inactivating T lymphocytes or natural killer cells or promoting the differentiation of regulatory T lymphocytes or myeloid cells to suppress immune responses [10]. Thomson et al. proposed that immature dendritic cells could be activated into immune-competent cells by binding and endocytosis of exosomes from other activated dendritic cells [11]. Yang et al. noticed that tumor cells could cause exocytosis anti-tumor drugs by EVs. Therefore, monitoring EVs can help us decipher a number of unclear physiological processes, including how tumor cells deceive immune cells, how cells collaborate, how cells react to the therapeutic treatments, and so on. It is meaningful for physiology, oncology, pharmacology, and many other disciplines to shed light on EV functions. EVs also play an important role in the tumor microenvironment formation and progression. Several studies have also been published examining the role of EVs in the different types of tumor models. Ji et al. found that ITGBL1-rich EVs were secreted by primary tumors to advance distal metastatic tumor growth [12]. Fang et al. suggested that intercellular crosstalk between tumor cells and fibroblasts was mediated by tumor-derived EVs that controlled lung metastasis of liver cancer [13]. Luga et al. showed an intercellular communication pathway whereby fibroblast EVs mobilized autocrine Wnt-PCP signaling to drive breast cancer cell invasive behavior [14]. Atay et al. revealed that EVs’ release and subsequent MMP1 induction promoted the invasion of gastrointestinal stromal tumor cells [15]. These are just a few examples out of hundreds that illustrated intriguing roles of EVs in tumor progression and metastasis. Furthermore, an increasing number of research studies have been conducted to demonstrate that EVs have other significant biological functions and can mediate many biological processes, such as angiogenesis and cell survival [16,17].

Accordingly, the diagnostic and prognostic value of EVs has been shown in multiple types of cancers [18]. The wide application of EVs can promote the use of EVs in body fluids as clinically useful diagnostic biomarkers and therapeutic targets for cancer, which may improve cancer management in the future [19].

Several strategies have been proposed to isolate EVs, including ultracentrifugation (UC), density gradient centrifugation, immunoaffinity interaction, precipitation-based aggregation, ultrafiltration, size exclusion, and so on [20,21]. UC, the current “gold-standard” technique, is used to separate and concentrate EVs from other specimen constituents according to their density. However, this method may isolate similarly sized particles and damage EV membranes, even inducing aggregation [22,23]. The low-throughput and high cost of UC may also not meet the standards required for clinical applications, thus limiting the clinical utility of this approach. The density gradient centrifugation method with an additional purification step can increase the EVs purity but results in a lower yield [24]. Immunoaffinity interaction strategies can reduce non-targeted EVs and microparticles by using antibody-antigen interactions, which are highly specific but costly, and has the drawbacks of nonspecific binding, competitive inhibition, cross-reactivity, and short lifetime of antibodies [25,26]. Other emerging approaches for particular specialized applications have been introduced in recent years but with limited success [27]. The inherent limitations of the other methods, such as substandard achievable purity, low yield, vesicle integrity, lengthy processing time, high cost, and poor recovery efficiency, restrict their potential applications for rapid, efficient, and convenient EV capture and downstream and analysis. Recently, several novel methods based on affinity nanoscience have demonstrated the ability to solve these limitations and shorten the effective handling time. Chemical affinity-based solid-phase isolation, such as TiO_2_ microspheres [28] or amphiphilic magnetic beads [29], allows faster capture of EVs with higher efficiency than UC. Such a more simplified process enables the preferred automated, streamlined capture and analysis of EVs.

To address the urgent needs, we introduce here a magnetic bead-based high-throughput platform, termed EVrich, for the isolation of EVs from urine while it is conceivable the system can be applied to other biofluids. We demonstrate its application in prostate cancer biomarker discovery. The platform includes the EV capture, washing, and elution steps in the automated and paralleled mode. We optimized these steps based on EVtrap beads [29,30], although it is conceivable that the platform is applicable to any magnetic bead-based isolation. The results were compared to those obtained by ultracentrifugation and manual EVtrap-based isolation for contamination levels, efficiency, and reproducibility of EV isolation. The EVrich approach demonstrated expected outcomes in all of the tested parameters. We showed that the EVs isolated by EVrich could subsequently be subjected to direct on-bead analysis or recovered in the elution solution for EV characterization, such as RPS, TEM, and WB, and cargo analysis of miRNAs and proteins (Figure 1). Finally, we demonstrated the application of EVrich with clinical samples by analyzing a cohort of prostate cancer samples and successfully quantified multiple EV miRNAs, proteins, and phosphoproteins specific to the disease. Collectively, the platform provides an attractive and simple strategy for routine handling of clinical samples for EV studies, facilitating the translation of EV-based biology and research to clinical applications with minimal hands-on time and high throughput.

## 2. Materials and Methods

### 2.1. Materials and Reagents

EVtrap^TM^ beads were provided by Tymora Analytical Operations (West Lafayette, IN). The automated magnetic bead separation and corresponding “96-well plate” of the EVrich system was in-house built. Triethylamine was obtained from Millipore-Sigma, and phosphotungstic acid was from Sinopharm Chemical Reagent Co., Ltd. DI water used in the experiments was prepared in a Milli-Q system (resistivity > 18.2 MΩ·cm). Lithiumdodecyl sulfate (LDS) loading buffer, tritonX-100 (Shanghai (China) Biotechnology Engineering Co., Ltd.), phosphate-buffered saline (PBS, Gibco), 15% sodium dodecyl sulfate-polyacrylamide gel electrophoresis (SDS-PAGE, Shanghai (China) EpiZyme Biotechnology Co., Ltd.), anti-CD9 (13403S, CST), anti-CD81 (ab79559, Abcam), anti-TSG101 (ab125011, Abcam), HSP70 (66183, Proteintech), Calnexin (ab133615, Abcam) and polyvinylidene fluoride (PVDF) membranes (Millipore, Sigma) were used in EV capture and western blotting experiment. Sodium deoxycholate, sodium lauroylsarcosinate, Tris (2-carboxyethyl) phosphine, 2-Chloroacetamide, and phosphatase inhibitor cocktail were all purchased from Millipore-Sigma to prepare EV lysis solution. Pierce™ BCA Protein Assay Kit was purchased from Thermo Scientific (23250, reducing agent compatible).

### 2.2. Collection and Preparation of Urine Samples

All clinical urine samples were obtained under approval from Jiangsu Province Hospital, Nanjing, China. All samples were collected with patients’ consent and used according to the ethical guidelines of the local hospital. On the first morning, urine samples were collected from healthy individuals, prostate cancer patients, and other prostatitis or prostatosis patients, followed by centrifugation at 3000× *g* for 10 min twice to remove cell debris and large apoptotic bodies. The final supernatant was stored at −80 °C for the subsequent operations.

For proteomics, urine samples with provided 10x Urine/Media Loading Buffer were added to the 96-well plate directly, incubated with magnetic EVtrap beads at 26 °C for 1 h to capture the EVs, and then the automated magnetic beads separation step started. As a result, EVrich was exploited to fully automate the process starting from cell or body fluid to EVs ready for downstream analyses for 48 samples in parallel.

### 2.3. Purification and Capture of EV Samples

The method of manually extracting EVs using EVtrap beads refers to the previous protocol [29]. After magnetic capture, EVtrap beads were washed twice with PBS for 10 min in total, and the EVs were eluted through incubation with 200 μL of 100 mM trimethylamine for 10 min. The EV extraction and purification processes by the automated instrument are shown in Figure 1A. The processes of EV proteomics, phosphoproteomics, and miRNA analysis are shown in Figure 1B. Instead of shaking, the approach of incubation was replaced by the shear stress mixing. After mixing 200 times in the first column and washing with the provided wash buffer once, the EVtrap beads were transferred to the next two columns and washed with PBS twice. Finally, the EVs were eluted with triethylamine twice in the last two columns separately. Purified EVs were dried under vacuum and stored at −80 °C until used.

### 2.4. Characterization of EVs Using Transmission Electron Microscopy

EVs corresponding to 1 mL urine sample were dispersed in 200 μL PBS to prepare EV-TEM solution. For each sample, 10 μL EV-TEM solution was dropped onto a 200-mesh formvar carbon-coated copper grid. After natural drying, the sample was incubated with 2% phosphotungstic acid solution (pH = 7.0) for 2–3 min at room temperature for negative staining. Ultimately, the transmission electron microscopy (TEM) images of EVs were carried out on a HITACHI H-8100 electron microscope (Hitachi, Tokyo, Japan).

### 2.5. Resistive Pulse Sensing (RPS) Analysis

The resistive pulse sensing (RPS) strategy, the so-called Coulter principle, has been extensively exploited in cell sorting and counting since the 1950s. Benefitted from modern micro and nano fabrication techniques, several powerful and effective methods and systems have been applied for sensitive, high-throughput, and high-definition characterizations of sub-micro scale objects, such as nanoparticles [31], viruses [32], and DNA [33]. Here, EVs corresponding to 1 mL urine sample were eluted twice with 200 μL of 100 mM TEA to form dry powder for storage and diluted to 0.5 mL with PBS for further RPS analysis. The size distribution of EVs captured by manual operation and by EVrich were determined using a Nanocoulter counter (Resuntech, co., LTD, Shenzhen) equipped with nanopore chips having a measuring range of 60–200 nm or 150–500 nm.

### 2.6. Western Blotting Analysis

EVs corresponding to 1 mL urine sample were lysed with 20 μL LDS loading buffer to gain protein lysates, followed by boiling at 95 °C for 5 min. Then, these samples captured by manual operation and EVrich were loaded onto a 15% SDS-PAGE (voltage 190 V, 70 min) in an equal-sample-volume way and transferred (current 275 mA, 70 min) onto PVDF membranes. Afterward, the PVDF membranes were blocked with 1% BSA in TBST for 1 h and incubated with anti-CD9, anti-CD81, anti-TSG101, anti-HSP70, and anti-Calnexinovernight in TBST containing 1% BSA at 4 °C. Followed by specific binding with HRP-conjugated secondary antibody in 1% BSA in TBST, the membranes were scanned with an enhanced chemiluminescence imager (ImageQuant LAS500) to achieve the immunoassay and quantitation. The test was repeated three times, and the averages were reported.

### 2.7. Samples Preparation of EVs for qRT-PCR

Total RNA from EVs was extracted using miRNA Purification Kit (CWbiotech, China) according to the instructions. Total RNA was reverse-transcribed into cDNA using miRNA 1st Strand cDNA Synthesis Kit (by stem-loop) (Vazyme, China). The qRT-PCR (quantitative real-time PCR) reactions were performed by CFX connect (Bio-Rad) with the miRNA Universal SYBR qPCR Master Mix (Vazyme, China). qRT-PCR reactions were run in triplicate. The primers used for RT-PCR (reverse-transcribed) and qRT-PCR were synthesized by Sangon Biotech and provided in Appendix A.

### 2.8. Preparation of EV Samples for LC-MS

First, EVs captured by UC, manual EVtrap, or EVrich methods were dissolved in a solution containing 12 mM sodium deoxycholate, 12 mM sodium lauroylsarcosinate, 10 mM Tris (2-carboxyethyl)phosphine, 40 mM 2- chloroacetamide mixture of 50 mM Tris HCl (pH 8.5) in lysis solution of phosphatase inhibitor cocktail. They were then boiled in a 95 °C water bath for 10 min and diluted five-fold with 50 mM triethylammonium bicarbonate. A 2.5% solution was removed for BCA assay to determine total protein content, and the remaining solution was digested with Lys-C (EVLiXiR) at 1:100 (*w*/*w*) for 3 h at 37 °C. Afterwards, the mixture was incubated with trypsin at 1:50 (*w*/*w*) overnight in a 37 °C water bath to further digest the peptides. Then, 10% trifluoroacetic acid (TFA) was added to acidify the sample to a final concentration of 1% TFA, and ethyl acetate was added to the above mixture to dilute it by a factor of two. The resulting solution was vortexed for 2 min and centrifuged at 15,000× *g* for 3 min. The upper organic phase was removed, and the lower aqueous phase was collected and lyophilized in a refrigerated vacuum centrifuge (Laconco CentriVap). The desalting process was performed using an 8 mm Capture Disk (3 M Empore 2240-SDB-XC) according to the manufacturer’s instructions. We removed 2% of each sample for direct proteomics experiments; the remaining 98% of each sample was used for phosphopeptide enrichment by the PolyMAC Phosphopeptide Enrichment Kit (Tymora Analytical) according to the manufacturer’s instructions and analyzed by a 60 min LC-MS run. All samples were freeze-dried in a refrigerated vacuum centrifuge and stored at −80 °C.

### 2.9. LC−MS/MS Analysis

The mobile phase buffer consisted of buffer A (0.1% FA in ultrapure water) and buffer B (80%ACN/0.1% FA). All peptides for proteomic experiments were separated on a 25 cm in-house packed column (360 µm OD × 150 µm ID) containing C18 resin (2.2 µm, 100 Å; Michrom Bioresources) at a flow rate of 600 nL/min with a linear 80 min gradient from 5% to 40% B, followed by a 10 min washing gradient. Phosphopeptides were analyzed at a flow rate of 600 nL/min with a linear 52 min gradient from 8% to 32% B, followed by an 8 min washing gradient. The EASY-nLC 1200 was coupled online with a Q Exactive HF-X mass spectrometry (Thermo Fisher Scientific, Bremen, Germany). The instrument was freshly cleaned and calibrated using Tune (version 2.9) instrument control software. All data were acquired using data-dependent acquisition (DDA) in profile mode using positive polarity. In this mode, the instrument automatically selected the “top-20” most abundant precursor ions from a full MS spectrum for subsequent MS/MS fragment analysis. Spray voltage was set to 2.1 kV, funnel RF level at 40, and heated capillary at 320 °C. Full MS resolutions were set to 60,000 at *m*/*z* 200, and the full MS AGC target was 3 × 10^6^3E6 with a maximum inject time (IT) of 30 ms. Mass range was set to 400–1200. The AGC target value for fragment spectra was set at 1E5, and the resolution threshold was kept at 3E5 with an IT of 50 ms. Isolation width was set at 1.6 *m*/*z*. The normalized collision energy was set at 28%. Only precursors charged between +2 and +7 that achieved a minimum AGC of 8 × 10^3^ were acquired. Dynamic exclusion was set to 40 s to exclude all isotopes in a cluster.

### 2.10. LC−MS/MS Data Processing and Quantitation

Thermo RAW files were processed using PEAKS Studio X+ software (Bioinformatics Solutions Inc.). The search was performed against the human UniProt database version downloaded in May 2020 with no redundant entries. The enzyme was set to trypsin/P with up to 3 missed cleavages. Carbamidomethylation (C) was selected as a fixed modification, while oxidation (M) and acetylation (protein N-term) were selected as variable modifications. The variable modifications of phosphorylation (S/T/Y) were also selected for the phosphopeptide sample search. The false discovery rates (FDRs) of proteins, peptides, and phosphopeptides were all set to 1% (−10 lgP ≥ 20 ≥ 1 unique peptide for proteins).

For the data processing of both proteomic and phosphoproteomic quantification, the intensities of peptides and phosphopeptides were extracted with initial precursor mass tolerance set at 20 ppm and PSM confidence FDR of 1%. All of the data were normalized using Total Ion Current (TIC) signals. The individual protein intensity was calculated by the top three peptides with the highest abundance. For calculation of fold changes between the groups of proteins, a label-free quantitative method was used to compare phosphoproteome differential expression, EV markers, and free urine proteins within different samples. Based on these results, the differential expression analysis of phosphopeptides in clinical samples was conducted using Perseus software for missing value replacement, normalization, and z-score normalization. The volcano plots and heatmaps (*p*-value < 0.05, t-test S0 = 0, log2 (Fold change) > 1 was regarded as differential proteins or phosphopeptides) were generated with R (version 4.0.5).

## 3. Results and Discussion

### 3.1. Implementing High-Throughput, Automated EV Isolation

Previous studies have demonstrated that EVtrap beads can efficiently extract EVs from different biofluids with high yield and purity and enable the efficient identification of a large number of EV-derived phosphorylated peptides [29,30]. Here, we adapted the EVtrap beads to a 96-well plate format with automated magnetic bead separation, which we name it EVrich, although it is conceivable the system can be applied to any magnetic beads-based isolation. The in-house system requires the optimization of multiple steps, including the amount of reagents, the concentration and volume of washing buffers and elution buffers, as well as optimizing the accessories for automatic magnetic bead separation, such as magnetic bars, magnetic sleeves, and the 96-well plate. In addition, optimization of the mixing speed, mixing time, magnetization time, and height of the magnetic sleeve at different steps to obtain the optimum extraction results. The isolation process, termed EVrich, is shown in Figure 1. On the prototype platform, EVs were extracted from 48 samples simultaneously using EVtrap beads with three 96-well plates.

To maximize the efficiency of the automated EV capturing process, we used healthy donors’ urine samples to optimize EVrich with regard to several parameters in the isolation procedure, including the initial amount of EVtrap beads, the incubation time, and washing/elution speed. All of the data were normalized to the highest data point in each group. The optimization process results are displayed in Appendix A. As shown in Appendix A, for each mixed urine sample, in the beginning, more EVtrap beads led to more isolation of EVs based on the signal of EV marker CD9, and the relative intensity of CD9 becomes almost constant after the concentration of EVtrap beads reaches 20 μL/mL of urine, demonstrating an adequate amount of EVtrap beads at this beads/urine ratio. Next, we examined the optimum incubation time between EVtrap and urine samples. As shown in Appendix A, it was difficult to capture EVs completely from samples within 20 min. Meanwhile, the relative intensity of the CD9 protein did not change significantly when incubating for more than 30 min. In the washing step, processing at excessive speed led to a devastating loss of protein (Appendix A). By multiple testing, the optimum washing speed was set as 100 mm/s. Finally, the elution speed was set to the maximum speed allowed, 650 mm/s (Appendix A). 

To evaluate the performance of EVrich relative to the existing methods for EVs capture, we performed EV isolation and characterization by automated EVrich, EVtrap manual, and UC. First, according to the TEM images (Figure 2A,B), the EVs isolated by the automated EVrich process demonstrated good size integrity, ranged in diameter from 50 to 150 nm, and showed a uniform spherical structure with a few impurities. Samples exhibited typical characteristics of extracellular vesicles [20]. Figure 2C shows the RPS results of the three EV samples collected by manual EVtrap, automated Evrich, and UC. The result shows that EVrich might extract more small size (under 200 nm) EVs than manual or UC. According to the results from EVrich extraction, the number of particles below 200 nm is basically 10 times that of 200–500 nm particles, which is much higher than that of manual extraction and UC samples (Appendix A). Where the result is interesting, the exact cause for isolating more small-sized EVs is unclear and requires further investigation. Subsequently, the presence of the common EV marker TSG101, CD9, CD81, HSP70, and the absence of negative control Calnexin were validated by performing western blot analysis (Figure 2D). The band intensity from Evrich isolation was similar to that of the manual group but significantly higher than that of the UC group. The smaller intragroup variability of the TSG101, CD9, CD81, and HSP70intensity in the automated group indicates the reliability and stability of the EVrich capture process. Calnexin is an endoplasmic reticulum protein, and Calnexin was not detected by the three methods, indicating that there was no protein contamination from cells in the extracted samples. In summary, the automated EV capture provides greater capture efficiency than UC while showing greater reproducibility and smaller variation than the manual operation.

To further demonstrate its application with clinical samples such as urine, we collected urine samples from three healthy donors twice a day for two consecutive days to confirm the stability of yield and purity when samples were loaded in different locations in the EVrich instrument (randomly selected) or processed on different days (Figure 3A). The WB results (Figure 3B,C) show remarkably consistent CD9 signals on different days with three individuals. Since the western blotting experiment could only be used to analyze differences in a few known EV markers between samples, we further applied LC-MS/MS to examine the whole EV proteomes for comparison. EV samples collected by ultracentrifugation and manual and automated capture were treated as described in the methods section to obtain peptides. The LC-MS/MS results showed that the signal intensities of the EV-specific proteins identified in manual and automated extractions were much higher than that after ultracentrifugation, while the signal intensities of representative contaminant proteins in urine were lower in manual and automated capture compared to UC (Figure 4A). This was basically consistent with our previously published results [29]. Data obtained using the automated EVrich extraction were basically consistent with the manual EVtrap results. As shown in Figure 4B,C, we identified on average 14,600 unique peptides corresponding to ~2095 unique proteins in the manual capture group, which was quite similar to automated capture (on the average of 2074 proteins) and significantly higher than after ultracentrifugation (about 1390 proteins). In addition, we compared the identified proteins with the Top 100 exosome proteins published on ExoCarta [34]. As illustrated in Appendix A, 87 and 88 proteins of the Top 100 exosome proteins were detected in samples after manual and automated operation, while only 85 proteins were detected in ultracentrifugation samples, indicating the high capture efficiency of the EVrich approach. The EV proteomics results extracted by the three schemes are shown in Appendix A.

To evaluate another major component in EVs, miRNA, after capture with the above three processes (UC, manual and automation), we focused on several common EV miRNAs species previously reported by other groups, miR-21, miR-125b and miR-221. MiR-21 has attracted the attention of researchers in various fields, such as development, oncology, stem cell biology and aging, becoming one of the most studied miRNAs [35]. MiR-125b was implicated to have close relationship with cell proliferation and differentiation, and downregulation of miR-125b was observed in various types of cancers [36]. MiR-221, located on human chromosome X, is upregulated in many different cancers. As an oncomiR or oncosuppressor-miR, MiR-221 plays an important role in tumor progression [37]. As shown in Figure 4D, the miRNA levels obtained through automated and manual extraction were similar. However, for all three miRNAs analyzed, the miRNA amounts after automated capture were 7-, 5- and 2-fold higher than using the ultracentrifugation method, respectively, indicating the superior performance of the EVrich method.

### 3.2. Application of EVrich for Prostate Cancer Biomarker Discovery

We next aimed to test the in-house EVrich system by applying it to a clinical application by processing a cohort of clinical samples. Protein phosphorylation is vital for cancer onset, progression, and drug resistance. Our previous studies have demonstrated that phosphorylated proteins in EVs might serve as diagnostic and prognostic biomarkers for breast and kidney cancers [30,38]. With the help of EVtrap beads and the EVrich system, we can quickly and effectively screen for phosphorylated protein biomarkers in urine EVs. Here, we selected urines from patients with prostatitis or prostatosis (control) and prostate cancer, which accounts for 14.1% of all cancers newly diagnosed [39], as a model to develop a liquid biopsy assay based on urine EVs and EVrich isolation.

To apply our automated EV capture process in practice, we collected urine samples from patients with prostate cancer (*n* = 21, Gleason score above 6) and with prostatitis or prostatosis as control participants (*n* = 21) (clinical information is available in Appendix A). Every 3 samples belonging to the same category were pooled as individual sample groups to get a total of 14 sample groups. All of the sample groups were processed with the EVrich procedure using the EVtrap beads, as shown in Appendix A. The isolated EV samples were lysed using the phase-transfer surfactants-aided procedure, followed by detergent removal, digestion, and desalting to obtain EV peptides [40]. About 2% of each peptide sample was directly used for proteome analysis. The remaining samples were further subject to PolyMAC phosphopeptide enrichment prior to LC-MS analysis [41]. Both proteome and phosphoproteome fractions were analyzed on a Q Exactive HF-X mass spectrometer coupled to an nanoEASY 1200 UHPLC [42]. Additionally, an indexed Retention Time Standard mixture (iRT) containing 11 artificial synthetic peptides was spiked into each sample and utilized as an internal standard to reduce run-to-run variations and assist peptide quantitation [43].

In total, we identified 47,563 unique peptides corresponding to 4328 unique proteins (Appendix A) and 8961 phosphorylated peptides representing 1606 unique phosphoproteins (Appendix A). Principle component analysis (PCA) on the EV proteins and phosphoproteins clearly separated the cancer groups from the control group (Appendix A). Further, in-depth data analysis was employed to obtain statistical results and generate volcano plots and visualized heatmaps (Figure 5A–D). In comparison to prostatitis or prostatosis samples, the prostate cancer groups identified 268 overexpressed proteins and 186 overexpressed phosphorylated peptides corresponding to 48 phosphoproteins, which were illustrated in the volcano plots (Figure 5A,C). The proteins and phosphopeptides included in the two heatmaps were listed in Appendix A. We enriched the differential proteins to 25 biological processes (*p*-value < 0.05), among which the biological process of cell adhesion, a process highly relevant to cancer invasion and metastasis, had the most enriched proteins and the highest confidence (Appendix A). Among the pathways enriched by differential proteins, the PI3K—Akt signaling pathway, which is associated with cancer, had the more identified proteins and higher confidence [44]. In addition, the confidence of the cell adhesion molecule pathway is also very high [45] (Appendix A). Among upregulated phosphoproteins, at least 14 of them have been previously reported to associate with prostate cancer: PLAU, SPP1, MCAM, CDH11, GDF15, IGSF8, CRISP3, LGALS3, FGFR1, HYAL1, and ENG. Our discovery of these proteins in urine EVs is quite appealing as these prostate cancer-related proteins were previously discovered in cell systems or animal models. Full validation of all these proteins will be beyond the scope of this study, and here we exemplify three proteins and one phosphoprotein (through quantification of its phosphopeptide) as potential biomarkers (Figure 5E–H). Previous studies indicated that secreted phosphoprotein 1 (SPP1) was involved in the recurrence and metastasis of prostate cancer [46] and regarded as a potential biomarker and therapeutic target for metastatic castration-resistant prostate cancer (mCRPC) [47]. Meanwhile, it had been shown that phosphorylated SPP1 secreted from cancer cells regulated cancer cell motility in human lung cancer [48]. In our data, we also found significant upregulation of SPP1 in the prostate cancer group (Figure 5E), and phosphorylation levels of SPP1 at 219 sites in the prostate cancer group were significantly higher than that in controls (Figure 5H). Importantly, a much larger difference in phosphorylated peptide at position 219 of SPP1 than in SPP1 quantitation indicates that up-regulation of SPP1 phosphorylation on position S219 in cancer patients is not only due to changes in SPP1 expression but also due to actual increase in phosphorylation on the specific site. Similarly, we observed that growth/differentiation factor-15 (GDF15) was highly expressed in EVs of the prostate cancer group (Figure 5F), which correlates with a previous study concluding that GDF15 can promote AR-positive prostate cancer cell proliferation through activating the ERK1/ERK2 signal pathway [49]. Finally, we found cell surface receptor protein ephrin A1 (EFNA1) is higher in prostate cancer EV than in the control group (Figure 5G). EFNA1 has been shown to be commonly overexpressed in malignant melanocytes and in prostate carcinoma cells (Figure 5G) [50]. Additional linear box-and-whiskers plots for other potential candidates are shown in Appendix A.

Taken together, with the use of EVtrap beads and the EVrich system, we were able to demonstrate a robust cancer biomarker discovery platform with good stability and efficiency, highlighting its potentially broad applications in future large-scale biomarker studies.

### 3.3. Analysis of miRNAs in EVs from Prostate Cancer Urine Samples

Finally, in addition to proteome analysis, we also used the EVs isolated by EVrich from prostate cancer patients and examined EV miRNAs as potential biomarkers for prostate cancer. The workflow is shown in Appendix A. MiRNA controls protein expression by degrading RNA or inhibiting protein translation, which affects a wide range of biological processes and is often deregulated in cancer. Multiple miRNAs in both cells and EVs have been reported to be involved in cell-to-cell communications and are promising molecules as disease biomarkers. For example, MiR-145 has been reported to be involved in the invasion and metastasis of prostate cancer in other studies [51,52,53]. As mentioned above, miR-125b was implicated in having a close relationship with cell proliferation and differentiation, and upregulation of miR-125b was observed in various types of cancers [36]. Previous Studies also have shown that miR-125b was highly expressed in the tissue cells of prostate cancer patients, while miR-145 was relatively low expressed [54,55,56]. The expression level of miR-125b and miR-145 both showed the same expression tendency as the previous reports, with the urine volume normalization of each sample to consist of a group in our study. Although some endogenous controls in exosomes have been reported, most of them were serum exosomal miRNA [57,58], and no urinary exosome endogenous controls related to prostate cancer have been reported. Therefore, in order to eliminate individual differences, we referred to the analysis method in EXO106 [59] to eliminate the differences in internal parameters by comparing EV miR-125b to miR-145 in each urine sample to get a score; we noticed that the score of prostate cancer group was significantly higher than the negative group (Appendix A). By comparing miR-125b to miR-145, we noticed that one parameter—the expression level of miR-125b relative to miR-145—could help us distinguish potential prostate cancer patients. 

## 4. Conclusions

To develop a high-throughput and reproducible EV isolation strategy, our lab designed an automated EVrich device that enables simultaneous separation and isolation of large batches of EV samples in parallel. By measuring the size distribution, yield, and purity of EVs extracted by EVtrap on the EVrich system, we demonstrated that the system can be applied to analyze a large number of clinical samples and enables the isolation of EVs potentially in clinical settings. Coupling with label-free proteomics and phosphoproteomics of urinary EVs from patients with prostate cancer and other non-cancer prostate conditions, we identified 268 significantly overexpressed proteins and 186 significantly overexpressed phosphorylated peptides corresponding to 48 proteins. Among them, multiple proteins and phosphoproteins were previously associated with either prostate cancer or other cancer types, indicating the success of the initial screening and urine EVs as the promising source for specific biomarkers for future exploration. In addition, we also demonstrated the isolation of miRNAs from EVs through the EVrich system by measuring miR-125b and miR-145, the ratio of which enabled statistical differentiation of prostate cancer. In conclusion, EVrich can achieve efficient EV extraction from dozens of samples in a relatively short period of time and minimize the technical variability caused by manual operation. This makes it suitable for large-scale EV-based clinical disease detection and opens new avenues for EV research and development and translational discovery in liquid biopsies.

## Figures and Tables

**Figure 1 cells-11-02070-f001:**
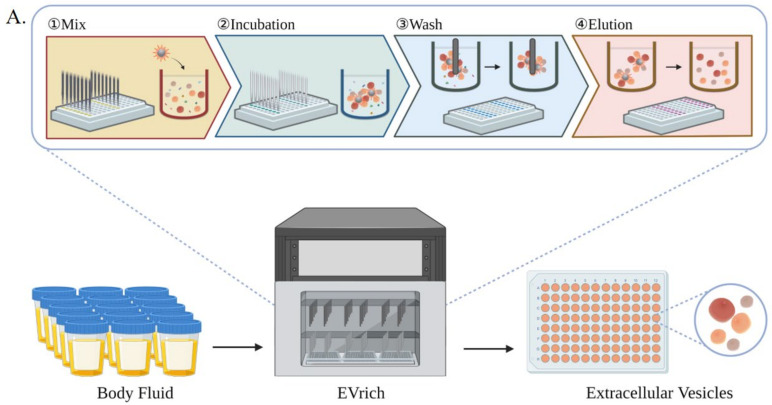
(**A**) schematic overview of the automated EV isolation workflow, including the incubation, washing, and elution steps. (**B**). Downstream EV analyses include EV characterization, EV proteomics, phosphoproteomics, and miRNA detection.

**Figure 2 cells-11-02070-f002:**
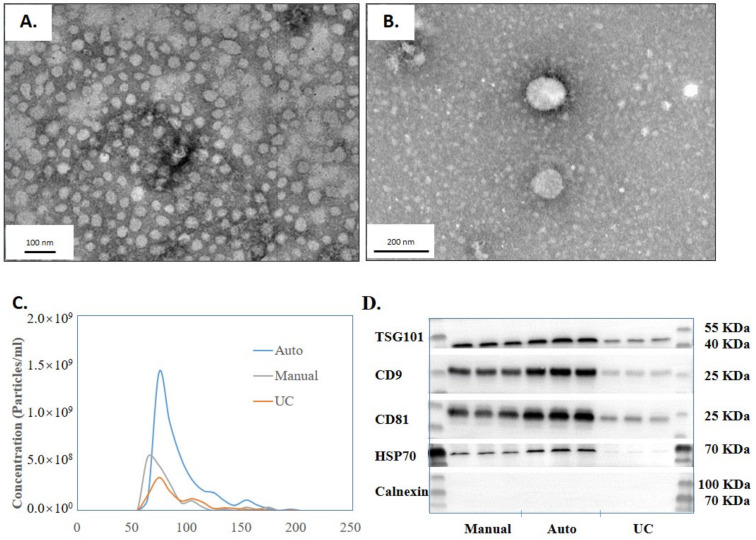
Characterization of the EVs isolated by EVrich. (**A**,**B**) The transmission electron microscopy characterization of the EVs. (**C**) RPS (Range 60 nm–200 nm) characterization of EVs isolated by the three methods. (**D**) Western blot detection of the CD9 protein content isolated by the three methods.

**Figure 3 cells-11-02070-f003:**
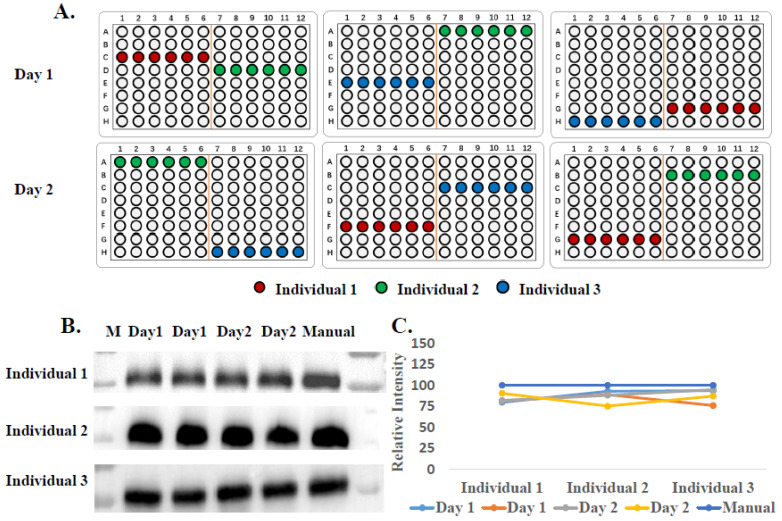
Stability assessment of the automated isolation. (**A**) The random sampling position sketch map of sample processing over two days. (**B**) Western blot analysis of CD9 signal from isolated EVs in urine. (**C**) Western blot CD9 quantitation from B.

**Figure 4 cells-11-02070-f004:**
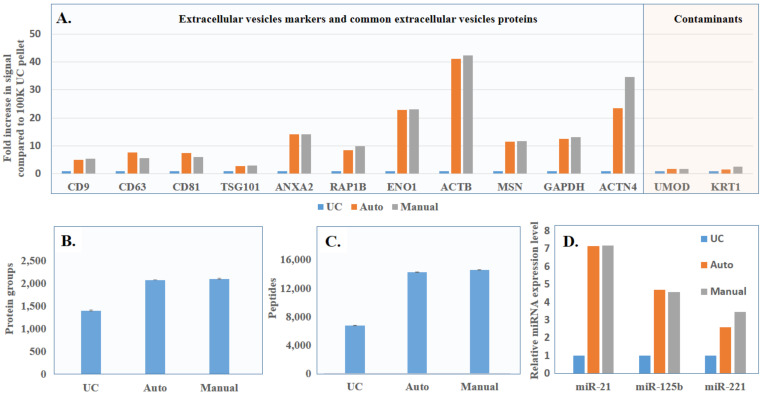
(**A**) LC-MS comparison of signal intensity of EV marker proteins and commons contaminants identified by the three isolation strategies. (**B**) LC-MS comparison of total EV proteins identified after isolation by the three different strategies. (**C**) LC-MS comparison of total EV peptides identified after isolation by the three different strategies. (**D**) Comparison of miRNA levels after isolation by the three different strategies.

**Figure 5 cells-11-02070-f005:**
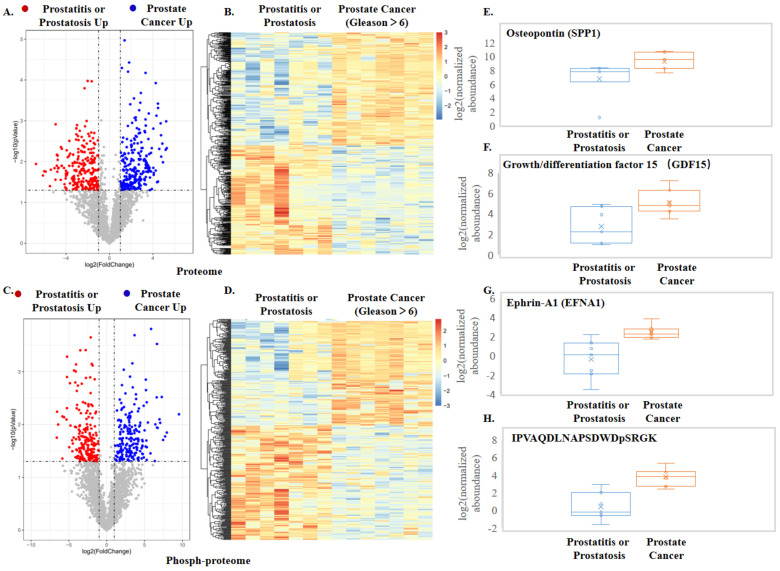
Results from quantitative proteomics and phosphoproteomics analyses of urine EVs from prostate cancer patients and prostatitis or prostatosis patients. (**A**) Volcano plot comparison of the regulated proteins. (**B**,**C**) Volcano plot comparison of the regulated phosphopeptides. (**D**) Heatmap of the significantly regulated overlapped phosphopeptides in the prostate cancer and control groups. (**E**–**G**) Quantitative measurement of prostate cancer-specific protein markers. (**H**) Quantitative measurement of the phosphopeptide in SPP1.

## Data Availability

MS raw data are available via ProteomeXchange with identifier PXD033595.

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
