# Peer review of "Proteomics, Phosphoproteomics and Mirna Analysis of Circulating Extracellular Vesicles through Automated and High-Throughput Isolation"

_cells, 2022, doi:10.3390/cells11132070_

Round 1

Reviewer 1 Report

Proteomics, phosphoproteomics and miRNA analysis of circulating extracellular vesicles through automated and high-throughput isolation

Interesting report by Zhang et al. Here the authors introduce a new method to enrich high yields of extracellular vesicles (EVs) easily and effectively from biofluids (urine). The issue addressed here is timely as the EV community would greatly benefit from such method. Unfortunately, here, lack of details surrounding the actual method (cf beads, automated instrument) and lack of controls greatly limit the impact of the present data. Further extensive work is needed before resubmission. See details below:

  • In general authors should refer to the ‘Minimal information for studies of extracellular vesicles 2018’ paper (MISEV2018) that compiles all requirements in terms of controls and validation experiments expected from any EV-focused studies, especially those that introduce new EV isolation methods. Authors should revise nomenclature and controls accordingly. For instance, based on the size of EVs they managed to isolate, it appears that the new method presented here is mostly able to enrich ‘small’ EVs, rather than bigger EVs such as ‘medium/large’ EVs or ‘oncosomes’ (cf MISEV2018). This is a limitation of their system that the authors need to address. In addition, WB validation of EV enrichment not only needs more EV specific markers (adding CD81, CD63 and HSP70 would be a good start), checking expression of contaminant markers is also necessary.
  • Most of references in the report are pre-2020. The EV field is a rapidly evolving field. Literature should be updated.
  • EV trap beads: main characteristics of these beads should be introduced. Illustrated description should be added to Figure 1.
  • Authors only tested processing of urine. It would be great to know how thei method performs with other biofluids such as blood or low input fluid such as CSF.
  • Line 170-173: not sure that type of information is relevant in the method section.
  • The only other real method the EVrich method is compared to here is ultracentrifugation (UC). Authors should at least compare their new method to another similar method (that includes magnetic bead-based enrichment) on the market. Alternatively, they could consider comparing their method to ‘combined’ methods, for instance UC combined to size exclusion chromatography.
  • Line 268: Authors mention that their system could be ‘applied to any magnetic beads-based isolation’ of EVs. Please mention a few and explain why that would be the case. It would be very interesting to see if authors can validate such statement by trying beads from different companies.
  • Figure 1 lacks information. There is no detail on the actual system (The instrument? How does it work?) provided here. The figure needs to have more complete legend that details the process.
  • Line 309: Authors mention a ‘few impurities’ in the TEM images. Please clarify.
  • Line 333-335: ‘EV-specific proteins’, which ones? Please clarify. Again, these would need to be validated by WB.
  • Line 337: ‘This was consistent with our previously published results’ = how’s that? Please clarify.
  • Line 343-346: Here there is one of many examples of overstatements that are present in the manuscript. Authors praise the ‘high capture efficiency of the EVrich approach’ although the difference reported here (3 proteins…) is far from being significant. Please tone down. Line 445-447 is another big overstatement. There is nothing in the present manuscript that shows evidence that the EVrich method is quicker (authors could time the different methods they tested and provide the data, for instance) or that the EVrich method allows to handle more samples at once (one can only assume this because of the 96w plate format but there is no actual proof that it’s the case).
  • Altogether it looks like the ‘manual’ and ‘automatic’ approaches perform in a similar extent for protein and miRNA identification and quantification even though the EV rich system produces way more EVs (cf Figure 2C): This raises questions re purity of the EVs enriched by the EVrich system. Is there any way the EVrich systems enrich contaminants/impurities within the EV size range? This should be addressed and discussed by the authors.
  • What’s the total amount of proteins obtained with each tested method?
  • Application of EVrich for prostate cancer biomarker discovery: this is great but how does this provide evidence that the EVrich method is effective, and performs better at describing new markers compared to other methods? There is no more comparison at this point. Any chance you can compare your data to available datasets from other teams?
  • Conclusion: line 479-480 and line 487-489 = for reasons mentioned above, there is unfortunately only limited proof in the present manuscript that this the case. Authors should tone down their conclusions or provide additional evidence.

Author Response

Interesting report by Zhang et al. Here the authors introduce a new method to enrich high yields of extracellular vesicles (EVs) easily and effectively from biofluids (urine). The issue addressed here is timely as the EV community would greatly benefit from such method. Unfortunately, here, lack of details surrounding the actual method (cf beads, automated instrument) and lack of controls greatly limit the impact of the present data. Further extensive work is needed before resubmission. See details below:

  1. In general authors should refer to the ‘Minimal information for studies of extracellular vesicles 2018’ paper (MISEV2018) that compiles all requirements in terms of controls and validation experiments expected from any EV-focused studies, especially those that introduce new EV isolation methods. Authors should revise nomenclature and controls accordingly. For instance, based on the size of EVs they managed to isolate, it appears that the new method presented here is mostly able to enrich ‘small’ EVs, rather than bigger EVs such as ‘medium/large’ EVs or ‘oncosomes’ (cf MISEV2018). This is a limitation of their system that the authors need to address.

Response: This manuscript introduces magnetic bead-based automation for EV isolation. In all experiments we used EVtrap as the example and restrictively speaking, this is not a new EV isolation method. In the manuscript, we characterized isolated EVs through TEM, NTA and WB. As suggested by the reviewer, we added a few more EV positive and negative markers in the WB experiments in the revision. The data were compared against manual procedure and ultracentrigution. The EVtrap-based isolation method, in theory, does not limit itself to small EVs but from the distribution of size range, the medium size is around 100nm so it is possible the method itself has certain biase. We revised the manuscript to address it as the reviewer suggested.

  1. In addition, WB validation of EV enrichment not only needs more EV specific markers (adding CD81, CD63 and HSP70 would be a good start), checking expression of contaminant markers is also necessary.

Response: As suggested by the reviewer, we added WB experiments on other EV positive and negative markers. As described in the manuscript, we used four positive controls and one negative control.

  1. Most of references in the report are pre-2020. The EV field is a rapidly evolving field. Literature should be updated.

Response: Updated as the reviewer suggested.

  1. EV trap beads: main characteristics of these beads should be introduced. Illustrated description should be added to Figure 1.

Response: EVtrap beads were introduced in 2018 and commercialized by Tymora Analytical Operations (West Lafayette, IN). They were fully characterized in previous studies (Ref. #22-23). EVTRAP enables the capture of EVs onto beads modified with a combination of hydrophilic and lipophilic groups that have a unique affinity toward lipid-encapsulated EVs.

  1. Authors only tested processing of urine. It would be great to know how thei method performs with other biofluids such as blood or low input fluid such as CSF.

Response: We indeed performed the EV isolation on the machine with other biofluids such as saliva and plasma. Each biofluid requires a slightly different modification on the procedure but we achieved similar good results. Not to confuse readers, we did not include the results with other biolfluids. We added one sentence in the revision accordingly.

  1. Line 170-173: not sure that type of information is relevant in the method section.

Response: This method is based on Coulter's principle, which is unmatched by any other nanoparticle analysis technique and is more accurate than commonly used light scattering techniques that provide bulk estimates. The Coulter principle consists of passing the suspended particles once through the orifice in the electric field of known characteristics; in the issue the momentary changes in impedance give rise to voltage pulses, the heights of which are proportional to volume (consequently, to volume diameter) of the particle traversing the orifice.

  1. The only other real method the EVrich method is compared to here is ultracentrifugation (UC). Authors should at least compare their new method to another similar method (that includes magnetic bead-based enrichment) on the market.Alternatively, they could consider comparing their method to ‘combined’ methods, for instance UC combined to size exclusion chromatography.

Response: In our study, we used EVtrap to demonstrate the automation and high throughput with the instrument. As EVtrap was already compared with a number of other methods in previous studies (Ref #22-23), we don’t feel it is necessary to make extensive comparison in this study.

  1. Line 268: Authors mention that their system could be ‘applied to any magnetic beads-based isolation’ of EVs.Please mention a few and explain why that would be the case. It would be very interesting to see if authors can validate such statement by trying beads from different companies.

Response: The capture method of this instrument is based on magnetic separation technology, and any reagent for EV isolation based on magnetic separation can use this device in principle. For example, ref #21introduced TiO2-based isolation of EVs and it is totally conceivable that the system can be applied if TiO2 has a manetic core.

  1. Figure 1 lacks information. There is no detail on the actual system (The instrument? How does it work?) provided here. The figure needs to have more complete legend that details the process.

Response: Figure 1 is the overview of the method and we included the optimization procedure on the instrument, such as incubation, washing, and elution steps, in Figure S1. To better serve the readers, we have added a video of the instrument running procedure in the Supplementary Material.

  1. Line 309: Authors mention a ‘few impurities’ in the TEM images. Please clarify.

Response: Corrected.

  1. Line 333-335: ‘EV-specific proteins’, which ones? Please clarify. Again, these would need to be validated by WB.

Response: The MS results identified multiple EV markers, such as CD9, CD63, CD81, TSG101, ANXA2, RAP1B, ENO1, ACTB, MSN, GAPDH and ACTN4, which are part of the top 100 exosome marker proteins published on ExoCarta (shown in Figure 4A). We have already validated the CD9, CD91, TSG101 by WB in this study.

  1. Line 337: ‘This was consistent with our previously published results’ = how’s that? Please clarify.

Response: We actually refer to the figure 3 of the paper “Highly Efficient Phosphoproteome Capture and Analysis from Urinary Extracellular Vesicles. J Proteome Res 2018, 17 (9), 3308-3316. DOI: 10.1021/acs.jproteome.8b00459.z”. We revised the manuscript to make it clear.

  1. Line 343-346: Here there is one of many examples of overstatements that are present in the manuscript. Authors praise the ‘high capture efficiency of the EVrich approach’ although the difference reported here (3 proteins…) is far from being significant. Please tone down.

Response: We have changed the description to tone down the statement as suggested.

  1. Line 445-447 is another big overstatement. There is nothing in the present manuscript that shows evidence that the EVrich method is quicker (authors could time the different methods they tested and provide the data, for instance) or that the EVrich method allows to handle more samples at once (one can only assume this because of the 96w plate format but there is no actual proof that it’s the case).

Response: We are sorry for not offering the time of each step. Here is the protocol with required time in each step. The total time is significantly shorter than UC and some other methods. We revised the manuscript to make it clear.

Stage

Time

Mix

30 min

Wash 1

3 min

Wash 2

3 min

Wash 3

3 min

Elution 1

3 min

Elution 2

3 min

Total

45 min

  1. Altogether it looks like the ‘manual’ and ‘automatic’ approaches perform in a similar extent for protein and miRNA identification and quantification even though the EV rich system produces way more EVs (cf Figure 2C): This raises questions re purity of the EVs enriched by the EVrich system. Is there any way the EVrich systems enrich contaminants/impurities within the EV size range? This should be addressed and discussed by the authors.

Response: The reviewer made a good point. Although it semes from Figure 2C the EVrich instrument isolated more EVs than the manual method, our BCA assays indicated similar extracted protein amount. The contamination levels are similar as well (Fig.4A). We will further explore the confusing result and at this stage, the result does not affect downstream analyses.

  1. What’s the total amount of proteins obtained with each tested method?

Response: Approximate 6ug of EV proteins can be obtained from 1 mL urine sample. We used 1 mL urine for typical proteomics experiments each time.

  1. Application of EVrich for prostate cancer biomarker discovery: this is great but how does this provide evidence that the EVrich method is effective, and performs better at describing new markers compared to other methods? There is no more comparison at this point. Any chance you can compare your data to available datasets from other teams?

Response: The reviewer made a good point but due to the scope of the study, we were not able to carry out side-by-side comparison on clinical samples using different isolation methods. We demonstrated good reproducibility, shorter time, and high throughput using the EVrich system through the actual application.

  1. Conclusion: line 479-480 and line 487-489 = for reasons mentioned above, there is unfortunately only limited proof in the present manuscript that this the case. Authors should tone down their conclusions or provide additional evidence.

Response: We have changed the description as suggested.

Reviewer 2 Report

The study is well designed and provides more detailed information in the field. I have a few suggestions for data analysis and presentation before accepting the manuscript. Please see the comments below.

Authors should provide the correlation plot among all the samples with correlation values.

It should be important to show the replicability of data among all sample types.

It will be valuable to describe the data in terms of a PCA plot for detailing the information of samples.

Proteomics and phosphoproteomics data should be used for the enrichment analysis and report the identification of the important terms identified. Based on this information, the authors can report on the signalling cascade and the critical biological process.

Minor comments:

Figure 2D reports the bar in the histogram. It is reported that what are these bars?

Figure 4 is presented with bar information. It would be beneficial if it was reported with bars.

Author Response

The study is well designed and provides more detailed information in the field. I have a few suggestions for data analysis and presentation before accepting the manuscript. Please see the comments below.

  1. Authors should provide the correlation plot among all the samples with correlation values.

Response: The reviewer made a good point. We actually made equivalent data analyses through the PCA plots (Fig.S5) and volcano plots, which revealed that the samples had larger differences between groups and much smaller differences within groups.

  1. It should be important to show the replicability of data among all sample types.

Response: We are not totally sure whether the reviewer meant the replicability using different biofluids or urine samples in different groups. For different biofluids, please also see our response to Reviewer 1’s question. We indeed performed the EV isolation on the machine with other biofluids such as saliva and plasma. Not to confuse readers, we did not include the results with other biofluids here. For urine samples from prostate cancer patients, we grouped them into different pools as biological replicates to show the replicability of data among all sample types.

  1. It will be valuable to describe the data in terms of a PCA plot for detailing the information of samples.

Response: The reviewer made great suggestion. We added the PCA plots of proteins and phosphoproteins in the Supplementary Figure S5.

  1. Proteomics and phosphoproteomics data should be used for the enrichment analysis and report the identification of the important terms identified. Based on this information, the authors can report on the signalling cascade and the critical biological process.

Response: The reviewer made a good point. We added the GO analysis and KEGG results, as shown in Supplementary Figure S7.

Minor comments:

  1. Figure 2D reports the bar in the histogram. It is reported that what are these bars?

Response: We had three technical replicates of western blot. We used image ananlysis software to read the band intensity of the target proteins and calculated their error bars of each group by the gray value.

  1. Figure 4 is presented with bar information. It would be beneficial if it was reported with bars.

Response: Thanks for the reviewer's suggestion. We changed the figure 4B and figure 4C in the revision.

Reviewer 3 Report

This manuscript by Zhang et al, aimed at developing a high throughput and reproducible platform for EV isolation from circulating biofluids based on a magnetic bead-based EV enrichment approach (EVrich). Further, the authors demonstrated that isolated EVs by EVrich could be used for downstream analyses such as: EV characterization, miRNA, and protein profiling. Despite the existence of several methods for EV isolation, EVrich represents a promising platform suitable for application in the clinical setting. Hence, this manuscript is very interesting. 

Nevertheless, the authors should consider the following points:

1. As one of the most used biofluid in the clinical setting is blood/plasma, it would be nice if the authors evaluate the performance of EVrich using plasma samples. 

2. Why did the authors choose CD9 to evaluate the distinct steps of the optimization process? Instead of using only one EV marker, the authors should combine distinct ones and evaluate the expression of negative control markers as well to further exclude the possibility of co-isolation of other organelles/lipoproteins…

3. The authors should clarify whether they have submitted the same urine sample to the distinct isolation methods. Further, it would be nice if the authors compare the performance of EVrich with SEC (size exclusion chromatography) as this method has been widely accepted and used for EV isolation.  

4. It is not clear which endogenous controls were used to normalize the expression levels of miRNAs. The authors should describe better in the material and methods section the analysis of miRNA quantification by qRT-PCR.

5. The legend of Figure 5 is not correct. 

6. Although it is beyond the scope of this manuscript, it would be nice if the authors validate some proteins by an independent method (WB for instance). 

Author Response

This manuscript by Zhang et al, aimed at developing a high throughput and reproducible platform for EV isolation from circulating biofluids based on a magnetic bead-based EV enrichment approach (EVrich). Further, the authors demonstrated that isolated EVs by EVrich could be used for downstream analyses such as: EV characterization, miRNA, and protein profiling. Despite the existence of several methods for EV isolation, EVrich represents a promising platform suitable for application in the clinical setting. Hence, this manuscript is very interesting. Nevertheless, the authors should consider the following points:

  1. As one of the most used biofluid in the clinical setting is blood/plasma, it would be nice if the authors evaluate the performance of EVrich using plasma samples.

Response: Please also see our previous response to Reviewer #1 on similar issue. We indeed performed the EV isolation on the machine with other biofluids such as saliva and plasma. Each biofluid requires a slightly different modification on the procedure but we achieved similar good results. Not to confuse readers, we did not include the results with other biolfluids. We added one sentence in the revision accordingly.

  1. Why did the authors choose CD9 to evaluate the distinct steps of the optimization process? Instead of using only one EV marker, the authors should combine distinct ones and evaluate the expression of negative control markers as well to further exclude the possibility of co-isolation of other organelles/lipoproteins…

Response: In the revision, we carried out additional WB experiments on both positive and negative EV markers (Fig. 2D in the revised manuscript).

  1. The authors should clarify whether they have submitted the same urine sample to the distinct isolation methods.Further, it would be nice if the authors compare the performance of EVrich with SEC (size exclusion chromatography) as this method has been widely accepted and used for EV isolation.

Response: In our study, we used EVtrap to demonstrate the automation and high throughput with the instrument. As EVtrap was already compared with a number of other methods in previous studies including SEC (Ref #22-23), we don’t feel it is necessary to make extensive comparison in this study.

  1. It is not clear which endogenous controls were used to normalize the expression levels of miRNAs. The authors should describe better in the material and methods section the analysis of miRNA quantification by qRT-PCR.

Response: We didn’t use any endogenous controls in this study. By comparing the ratio of up- against down-regulated miRNAs, we normalized the expression levels of miRNAs without the need to include endogenous controls, and this strategy has been proposed in multiple previous studies.

  1. The legend of Figure 5 is not correct.

Response: We are sorry for the mistake. The order of the figures was wrong. It has been corrected in the revised manuscript.

  1. Although it is beyond the scope of this manuscript, it would be nice if the authors validate some proteins by an independent method (WB for instance).

Response: Within the scope of this study as the reviewer pointed out, instead of cherrypicking several proteins for experimental validation, we compared our data and identified multiple EV proteins and phosphoproteins that were associated with prostate cancer in previous studies with tissue samples, animal models, and model cell systems.

Round 2

Reviewer 1 Report

Here is a revised version of the manuscript by Zhang et al. Overall authors addressed a lot of the issues reported at the time of the initial submission, including lack of controls and details, but also nomenclature. Yet, a few issues remain and still need to be addressed.   

-       There is still not much information on the type of instrument the authors used for performing the EV isolation. Authors mention they provide a video that is unfortunately not available in the supplementary material provided for peer-reviewing. Please clarify. If this new method is ‘universal, scalable, and simple’ as claimed by the authors in the abstract, then clear and detailed information on the instrument should be available in the present manuscript. Also Figure 1 could do with more specific labeling including timing and volumes.

-       Comparison of the introduced method to other known methods is still very limited. EVrich is only compared to UC here, which is only one of many other methods for EV isolation. The present EVrich method might be quicker than the other known methods but data presented here on the purity of the isolated EVs are too limited to draw proper conclusions. Also, there is no precise information on the costs of EVrich compared to other methods. Authors should address these very important points in the discussion. 

-       Line126-128 ‘The EVs captured…. (Sinopharm Chemical Reagent Co., Ltd).’ Is this information supposed to be in the ‘Material and reagents’ section? It seems like this should rather belong to the TEM section of the methods.

-       As mentioned at the time of the original submission, it is very unusual to see information found on line 173 to177 in the methods section. Please move to introduction if you think this is relevant information.

-       Figure 2: Blots in panel D are cut in the middle of the signal. Please edit. A positive control for Calnexin should be provided. Line 346-347: rewrite, Calnexin is not a common EV marker.

-       Line 439-440: ‘The quantification data of… (Figure S5).’ This is very vague and needs to be clarified. Explain how the mentioned data differentiate the cancer group from the control group?

-       Line 446-448: Explain how ‘the biological process of cell adhesion’ is relevant for differentiation between cancer and control groups. 

-       Line 513-517: Very confusing, please rewrite. 

-       In general, English language, including grammar and spelling, should be revised with help from a native English speaker. 

Reviewer 2 Report

The authors have sufficiently substituted the additional information. As a result, the work is improved, and the flow is also good. Additionally, there are no concerns since the authors have integrated and offered sufficient responses to the questions that were expressed. Therefore, following a thorough examination of the whole manuscript, I am satisfied with the current revision.

Author Response

We thank the reviewer's effort to improve our manuscript.